# WAREX: WEB AGENT RELIABILITY EVALUATION ON EXISTING BENCHMARKS

## ABSTRACT

Recent advances in browser-based LLM agents have shown promise for automating tasks ranging from simple form filling to hotel booking or online shopping. Current benchmarks measure agent performance in controlled environments, such as containers or stable networks, where websites behave deterministically. However, in the real world, users access websites over networks and HTTPS connections that introduce instability from multiple sources: client-side, server-side issues or broader system failures. Moreover, live websites are prone to web attacks such Cross-Site Scripting, as well as general site modifications which can cause unexpected or malicious pop-ups or improper functionality. To address this gap, we present WAREX[1], a plug-and-play, network-layer tool that integrates with existing web agent benchmarks by simulating common website failures. We measure the impact of WAREX across three popular benchmarks: WebArena, WebVoyager, and REAL. Our experiments show that introducing WAREX leads to significant drops in task success rates, highlighting the limited robustness of state-of-the-art agents. We demonstrate that WAREX serves as more than a diagnostic tool. By fine-tuning an open-source model (Qwen3-8B) on WAREX-generated "failure-recovery" trajectories, we achieve an 88.9% relative improvement in error recovery rates, validating WAREX as a core component for training the next generation of reliable web agents.

## 1 INTRODUCTION

*Web agents are leaving the lab and entering the wild, but benchmarks give a false sense of reliability.*

Web agents have emerged as a promising paradigm for automating complex online tasks, attracting significant attention across academia and industry. Recent advances have produced state-of-the-art web agents with diverse designs, ranging from variations in prompting and observation spaces to reinforcement learning-based action policies. Notable examples include SteP (Sodhi et al., 2024), WebNaviX (Shlomov et al., 2024), Agent Q (Putta et al., 2024), and GUI-Owl (Ye et al., 2025), among a myriad others. Large technology companies have also begun deploying production-grade agents, such as OpenAI (2025); Perplexity (2025) and TinyFish (2025). As these systems transition from research prototypes to real-world deployments, ensuring their robustness is ever more critical.

To assess progress, the community has introduced numerous benchmarks, including WebArena (Zhou et al., 2023), Mind2Web (Deng et al., 2023), WebVoyager (He et al., 2024), WorkArena (Drouin et al., 2024), WorkArena++ (Boisvert et al., 2025b), WebLINX (Lù et al., 2024), REAL (Garg et al., 2025), and AgentRewardBench (Lù et al., 2025), typically consisting of containerized environments and evaluation harnesses to measure success rates on browser-based tasks. While effective for assessing reasoning and action planning, they fall short of capturing the realities of operating on the open web. This is due to three critical simplifying assumptions that limit their validity in the wild. First, they assume a failure-free infrastructure, where agents interact with perfectly-functioning websites over a stable network. In practice, while performing a task on a website, agents (and humans as well) may encounter failures such as network delays, DNS outages, partial page loads, and transient client- or server-side errors—conditions that can derail task execution.

---

[1] **W**eb **A**gent **R**eliability **E**valuation on e**X**isting benchmarks. Named after the U.S. Army's WarriorExercise (**WAREX**), which immerses military units in realistic combat scenarios to prepare them for real deployment.

Second, they ignore adversarial manipulation. Recent studies reveal that deployed agents, such as Comet, can be compromised by hidden instructions embedded in page content (e.g., indirect prompt injection or cross-site scripting) (Brave, 2025), yet no benchmark tests for such vulnerabilities. Finally, many benchmarks are static and closed. They rely on frozen or simplified website snapshots, exhibiting deterministic behavior. This prevents evaluation under dynamic conditions such as evolving site layouts, personalized configurations, or feature rollouts. Closed benchmarks further restrict extensibility, making it impossible to test agents on new scenarios. Together, this leads to overly optimistic performance estimates and masks critical reliability gaps, an unacceptable risk as agents are deployed at scale (e.g., "running thousands of enterprise workflows per minute," as envisioned by TinyFish (2025)).

To bridge this gap, we introduce WAREX, a plug-and-play framework that augments existing benchmarks with realistic stress conditions. Rather than creating yet another benchmark, we design WAREX to act as a transparent proxy layer between agents and environments. By intercepting and modifying network traffic, it can inject (1) common web failures, (2) adversarial attacks, and (3) dynamic content variations, without altering the agent or benchmark source code. Enabling users to systematically evaluate robustness against such realistic failures using existing benchmarks. This novel design makes WAREX modular, benchmark-agnostic, and easy to integrate. Beyond robustness testing, WAREX enables efficiency analysis by logging LLM interactions, token usage, and latency—metrics often hidden in third-party agents. This dual capability provides a holistic view of both reliability and cost-effectiveness.

We validate WAREX on three widely used benchmarks, WebArena (Zhou et al., 2023), REAL (Garg et al., 2025), and WebVoyager (He et al., 2024), using their released agents. Despite strong performance under default settings, these agents exhibit severe degradation under conditions introduced by WAREX, exposing fundamental robustness gaps. We further explore mitigation strategies (e.g., prompting-based defenses) to illustrate how WAREX can guide future research. We demonstrate that training on WAREX-generated failure logs allows agents to improve their error recovery rate. Our results demonstrate that WAREX provides a principled, extensible framework for evaluating and training web agents under realistic, failure-prone environments, an essential step toward safe and reliable deployment.

## 2 RELATED WORK

**Safety Benchmarks**: With the growing interest in web agent development there has simultaneously been progress in benchmarks that assess their potential for misuse. These tend to focus on harmful tasks such as misinformation, illegal activity and social bias (Tur et al., 2025). Another rapidly developing area lies in LLM evaluation tools, assessing the safety of web agent actions (Yuan et al., 2024) or identifying risky tool use behaviors (Ruan et al., 2023). By contrast WAREX focuses on safety in deployment to the wilds of the web.

**LLM Attacks:** Another line of research focuses on robustness of LLMs in their uni- and multi-modal forms. Typically this is done by crafting adversarial inputs to induce harmful or unexpected behaviors (Shayegani et al., 2023) (Bailey et al., 2023). Most research is focused on standard deployments. However, robustness in web agents can prove significantly more challenging. These are multi-turn systems that may appear safe and reliable at one step and then fail at the next. For example, by entering sensitive user information in malicious websites or totally crashing due to a simple transient error. ST-WebAgentBench (Levy et al., 2024) introduces some policies for evaluating this form of robustness, but most existing and widely resourced benchmarks lack built-in harmful tasks or web states. Ideally, we want a solution that can take existing resources and turn them adversarial, providing an abundant testbed for assessing robustness.

**Web Agent Attacks:** The issue of web agent reliability represents a research frontier. Initial work by Zhao et al. (2023) demonstrated that agents frequently fall for malicious popups, highlighting a critical vulnerability to overlay-based attacks. Their methodology is wrapper-based: it alters the agent's rendered observations (screenshots and accessibility trees) and measures whether an agent clicks an injected overlay. This approach stops after analyzing the initial click; it cannot continue running the agent to see how it adapts and how task performance is affected. DoomArena (Boisvert et al., 2025a) similarly works at the application level. The authors inherit the environment from the benchmark and use a designed AttackConfig to modify the observation before it goes to the agent

which decides the next action. It can inject adversarial content into BrowserGym environments (Le Sellier De Chezelles et al., 2025) / $\tau$-bench based automation layers, allowing it to approximate DOM-level effects, measuring end-to-end task impact. Due to dependence on specific automation layer frameworks it is unsuitable for many popular benchmarks (e.g. He et al. (2024); Rawles et al. (2023; 2025); Kapoor et al. (2024)).

**WAREX:** In contrast to the application-level approaches above, WAREX is designed for maximum interoperability, operating at the network level via a transparent proxy. By intercepting and modifying HTTP(S) traffic, it genuinely alters the server response rather than just the agent's perception. Because these modifications happen at the network layer, WAREX requires *no changes* to benchmark or agent source code and is plug-and-play for *any benchmark/sandbox environment which runs over a network*, including ones where the source code is not public. Furthermore, WAREX expands the evaluation scope beyond adversarial overlays to include pervasive infrastructure failures, such as transient network, server errors or Javascript runtime delays. Finally, unlike tools that serve solely as diagnostics, WAREX functions as a remediation engine: its execution logs provide the "failure-recovery" trajectories necessary to train agents that are robust to these instabilities.

# 3 WAREX FRAMEWORK

## 3.1 HIGH-LEVEL ARCHITECTURE

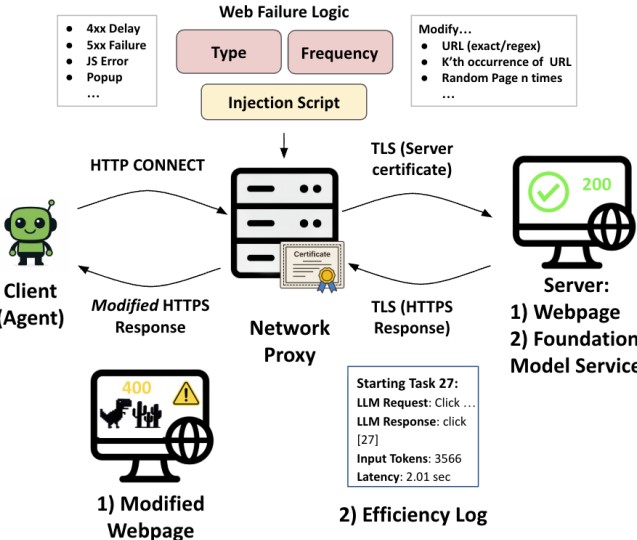

Figure 1: **WAREX Framework**. A network proxy splits TLS between client and server and runs an injection script with web failure logic: **Type** (failure mode — network delay, 5xx, JS error, popup) and **Frequency** (targeting injection policy — exact/regex URL(s); k'th/every-k/random n occurrences). The proxy rewrites selected responses and returns a modified page to the agent which it uses to decide its next action, while the server remains unchanged.

The high-level architecture of WAREX is shown in Figure 1. A client (web agent) issues a request to visit a website to execute a benchmark task; that request is routed through a proxy that implements the injection logic for controlled faults (e.g., delays, HTTP 4xx/5xx responses, JavaScript failures, popups/overlays) and terminates TLS. The proxy performs "split TLS": it holds one encrypted connection to the client (presenting an interception certificate) and a separate encrypted connection to the origin server, so it can decrypt client requests, apply Web Failure Logic, and re-encrypt traffic to the server. An Injection Script specifies the failure Type (e.g., 4xx network delay, 5xx server error, JS failure, popup/overlay) and the Frequency rules that determine when injections occur (e.g., exact URL or regex match, k-th occurrence of URL, first k occurrences, every k-th occurrence, or random selection n times). The proxy may forward the origin server's normal HTTPS response unchanged or rewrite it according to the Web Failure Logic; the agent observes the resulting Modified Webpage

and decides its next action accordingly, while the origin server and its content remain unchanged. WAREX also allows for the collection and logging of key performance metrics measuring agent efficiency through response latency, API call counts, input token, and output token counts. We describe different web failure injection types and details of our implementation of WAREX below.

## 3.2 Web Failure Types

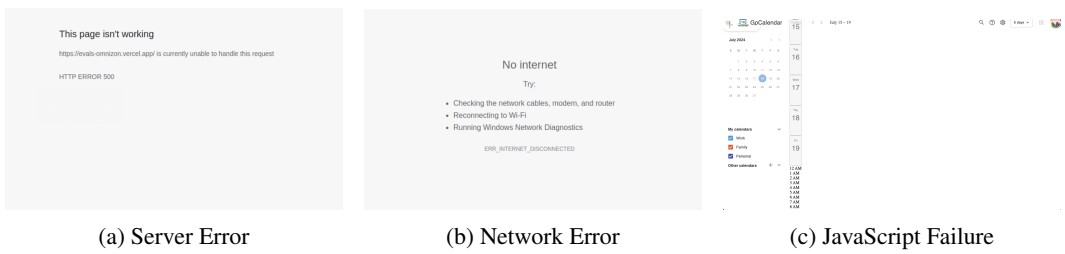

Figure 2: Default home page for GoCalendar task type in REAL benchmark with no fault injected.

(a) Server Error       (b) Network Error       (c) JavaScript Failure

Figure 3: Unreliable scenarios created on GoCalendar using the WAREX framework.

We experiment with these three common web failures, though there are more types that can be implemented using the WAREX framework. These failures are chosen based on prior empirical studies that highlight common failures experienced during accessing web sites (Padmanabhan et al., 2006; Singh, 2005; Ma & Tian, 2007; Ocariza Jr et al., 2011).

- **Network Errors**: These simulate client-side connectivity issues, such as slow loading, connection timeouts, which previous studies have shown to be common (Padmanabhan et al., 2006). WAREX introduces such issues by adding delays at the proxy level and displaying an error page instead of the normal page content. In our experiments we use 10 second delays. We expect the agent to refresh the page the way a human would.

- **Server-side Errors**: These represent temporary server-side failures that are common and are typically handled by retrying the request (Singh, 2005; Ma & Tian, 2007). This failure simulates HTTP error codes such as 408 (Request Timeout), 429 (Too Many Requests), 502 (Bad Gateway), and 503 (Service Unavailable). We show the common 500 error code in our experiments. We expect the agent not to timeout, but instead refresh the page and retry, as with the network error example, behaving constructively as a human would.

- **JavaScript Failures**: Motivated by prior studies on common JavaScript failures (Ocariza Jr et al., 2011), we simulate HTTP 504 (Gateway Timeout) errors, where crucial JavaScript functionality has not yet loaded. We add a 10 second delay to JS endpoints, causing certain images or buttons to appear broken or missing. A human user would notice the page has not loaded properly through its broken appearance and inoperative buttons. We expect the agent to detect this as well and refresh the page before proceeding with the task.

Figure 2 shows what an agent trying to browse the home page of the GoCalendar site from the REAL benchmark would see. In contrast, Figure 3 shows what the agent would see when the various web failures mentioned above are injected.

## 3.3 FAILURE FREQUENCIES

Users can configure the type as well as the frequency of injected failures in WAREX, creating a large space of potential stress-testing policies. As illustrated in Figure 1, users first define an exact or regex pattern match to target specific endpoint URLs. To control the timing of these injections, WAREX offers two primary behaviors. When the frequency parameter is set to zero, the system operates in a deterministic mode, intercepting only the first matching URL to test immediate recovery on a known page. Conversely, setting the frequency to a positive integer ($x > 0$) activates a stochastic injection strategy. In this mode, the proxy evaluates each matching request against a random probability; if selected, the failure is injected, and the injection counter is incremented. This process continues until the failure has been injected exactly $x$ times, allowing users to distribute a controlled number of faults unpredictably across a task trajectory to prevent agents from overfitting to fixed failure patterns. We have implemented example scripts to facilitate running the proxy with these various configurations.

## 3.4 TRAJECTORY LOGGING

In addition to intercepting webpage requests (Figure 1), WAREX can also capture calls to foundation model services that drive the agent's backbone and decision-making. These intercepted calls allow WAREX to record *efficiency metrics*, such as task latency, number of remote API calls, fine-grained details like LLM token counts, and precise logs of each LLM request-response pair. Network-level metrics (e.g., latency) are derived from request/response timestamps, while application-level metrics (e.g., token usage) are parsed directly from request contents. Some agents already measure and report such metrics. They implement logging/tracing by writing code inside the framework layer that wraps the model client. Other frameworks do not have this built-in functionality (e.g., SteP, REAL do not report LLM token counts per task). WAREX handles this and requires no modifications to the agent or benchmark. Instead, it treats the agent as a black box: a proxy with a coupled injection script logs all calls to specified endpoints. Developers can thus record detailed efficiency metrics even when the benchmark code is closed-source or inaccessible. Figure 1 demonstrates this architecture design.

## 3.5 SYSTEM IMPLEMENTATION

**Network proxy:** We use `mitmproxy`(Cortesi et al., 2010–), an open-source HTTP(S) proxy, to transparently capture network interactions and modify traffic in real-time. By default, Mitmproxy listens on port 8080 to intercept and process requests. We install Mitmproxy's built-in Certificate Authority (CA) as a trusted certificate within WAREX sandbox. This ensures that Mitmproxy can intercept, decrypt, and re-encrypt HTTP(S) traffic by acting as a "man in the middle" between any client (e.g., the agent) inside the sandbox and a server outside it. To introduce controlled, unreliable behavior into a benchmark, we leverage Mitmproxy's addon mechanism.[2] The addon mechanism allows injecting custom logic to hook into and modify Mitimproxy's behavior on how it forwards/blocks/manipulate traffic. WAREX addon for Mitmproxy operates in conjunction with a `config.json` file, which allows users to specify the types of unreliable conditions they wish to simulate.

**Sandbox:** Our prototype uses a Linux sandbox, though Mitmproxy can also be set up on MacOS and Windows. A key step is to configure the sandbox so that all network traffic from and to the agent is routed through WAREX proxy. Our implementation supports multiple mechanisms. (1) For Python agents, we set environment variables: `http_proxy: http://127.0.0.1:8080`, `https_proxy: http://127.0.0.1:8080` that ensures that network calls made by Python `requests` package go through the proxy. (2) Some agents run within their own sandboxes. For example, agents such as Stacked LLM Policy (SteP) Sodhi et al. (2024), use Playwright (Microsoft, 2020) to launch a Docker container (Docker Inc., 2013) as their execution environment. In such cases, we configure Playwright to use an explicit HTTP proxy by adding the following setting: `proxy={"server": "http://{your_server_hostname}:{port_number}"}` This routes all Playwright-driven network interactions, including remote service requests, through

---

[2]`https://docs.mitmproxy.org/stable/addons-overview/`

Mitmproxy. (3) For completeness, we also experimented with a third approach that routes all system-wide traffic through the proxy using Linux `iptables`. Since this method was not used in our experiments, we describe it in more detail in Appendix B.

# 4 EXPERIMENTS

## 4.1 EXPERIMENTAL SETUP

We use three web agent benchmarks: WebArena, REAL, and WebVoyager. Each benchmark includes a set of tasks to be performed on websites accessed via the network. We also define the corresponding agent used for each benchmark.

- **WebArena** (Zhou et al., 2023) features a Dockerized collection of synthetic yet realistic sites, such as a CMS store, a forum, GitLab, a map environment, and Wikipedia. We execute a total of 660 tasks, excluding those for the `Maps` environment due to setup issues. We used **SteP** (Sodhi et al., 2024) as the agent that interacts with websites via Playwright and provides the accessibility tree of the current webpage as its observation to the LLM.
- **REAL** (Garg et al., 2025) hosts Next.js replicas of popular websites like `Omnizon` (Amazon), `Udriver` (Uber), and `NetworkIn` (LinkedIn) on Vercel. Agents can access these websites over the open internet without any local setup. We utilize all 112 tasks. We used **REAL Demo Agent**, the reference "basic" agent. Like SteP, it operates browsers via Playwright and we provide the accessibility tree and a screenshot as its observation to the LLM.
- **WebVoyager** (He et al., 2024) comprises 643 tasks on 15 popular live websites, including Amazon, Google Maps, Wikipedia, and ESPN. We used **WebVoyager Agent**, the benchmark's default agent. It interacts with the browser via Selenium, providing the LLM with a screenshot and a simplified HTML representation of the current webpage as its observation.

The maximum number of steps per task is set to 20 for SteP, 25 for REAL, and 30 for the WebVoyager agent. All three benchmarks require an LLM backbone for deciding the next action based on the observation provided.

## 4.2 EXPERIMENTAL DESIGN

**Main Experiments:** We conduct the primary experiments using `GPT-4o` across all three benchmarks (WebArena, REAL, and WebVoyager). For each benchmark, we evaluate agents under seven conditions: 1) no proxy, 2) proxy without faults, 3) network error, 4) server error, 5) JavaScript delay 6) network error prompt improvement 7) server error prompt improvement. The results on WebArena are reported in Figure 4.

**Prompt-Based Mitigation and Backbone Comparison:** To improve agent robustness when faced with each web failure scenario, we explore a prompting mitigation strategy. For both server and network errors we guided the agent to refresh the page if faced with any errors (Appendix C). Extended results for prompting across all three benchmarks are shown in Table 1 in Appendix A. Rather than solely evaluating agents through success rate, we count the number of times the agent executes a recovery action (e.g., reload) in response to injected server, network, or JavaScript errors. If we express WAREX's logging structure for a given task as a trajectory $\tau = (s_t, f_{\text{inj}}, a_{\text{rec}})$, let $\mathcal{F}_{\text{inj}}$ be the set of all trajectories where a failure was injected, and let $\mathcal{R}_{\text{correct}} \subseteq \mathcal{F}_{\text{inj}}$ be the subset of those trajectories where the agent executed the valid recovery action. We define the Recovery Rate (RR) as:

$$\text{RR} = \frac{|\mathcal{R}_{\text{correct}}|}{|\mathcal{F}_{\text{inj}}|}$$

Since each agent framework relies heavily on the LLM backbone for decision-making, we investigate how different LLMs affect agent behavior. Beyond `GPT-4o`, we experiment with two additional models on REAL: `Qwen2.5-VL` (72B, open-source VLM) and `GPT-OSS` (20B, open-source, text-only). Results of these comparisons are reported in Figure 5.

324
325
326
327
328
329
330
331
332
333
334
335
336
337
338
339
340
341
342
343
344
345
346
347
348
349
350
351
352
353
354
355
356
357
358
359
360
361
362
363
364
365
366
367
368
369
370
371
372
373
374
375
376
377

**Failure-Recovery Training:** To demonstrate that WAREX functions not merely as an interception mechanism but as a core component of the agent development workflow, we designed a proof-of-concept fine-tuning experiment on REAL. We focused specifically on Server Errors, as the expected recovery action of reloading the page via 'goto(current_url)' is well-defined. To ensure the agent learns generalized recovery logic rather than memorizing specific steps, we configured WAREX to inject errors stochastically on random pages rather than consistently at the start of a task. We constructed a balanced dataset comprising equal parts "failure scenarios" (requiring a reload) and "clean runs" (requiring normal navigation). We partitioned the 11 websites within REAL into disjoint sets: 9 training, 1 validation, and 1 testing. This process yielded 244 traces for training, with 42 traces each for validation and testing. While this pilot dataset is relatively small, the methodology is designed to be extensible to larger models and diverse error types, such as network timeouts or popups. We selected Qwen3-8B-Base as the backbone model. Given the scale of the dataset, we employed Parameter-Efficient Fine-Tuning (PEFT) using Low-Rank Adaptation (LoRA) rather than full-parameter tuning. The model was trained via Supervised Fine-Tuning (SFT) for 4 epochs with a batch size of 8. We utilized a LoRA rank of 8 and alpha of 16, targeting all linear modules. The learning rate was set to $1 \times 10^{-4}$ with a cosine scheduler (0.5 cycles) and no warmup. To maintain stability, we applied a max gradient norm of 1. This setup allows us to isolate the impact of WAREX-generated data on the agent's ability to recover from fundamental web failures.

In addition to these three experiments, we explored malicious popup attacks, as they pose a unique challenge that closely mimics real-world adversarial threats (Appendix D).

## 5 RESULTS

### 5.1 MAIN EXPERIMENTS

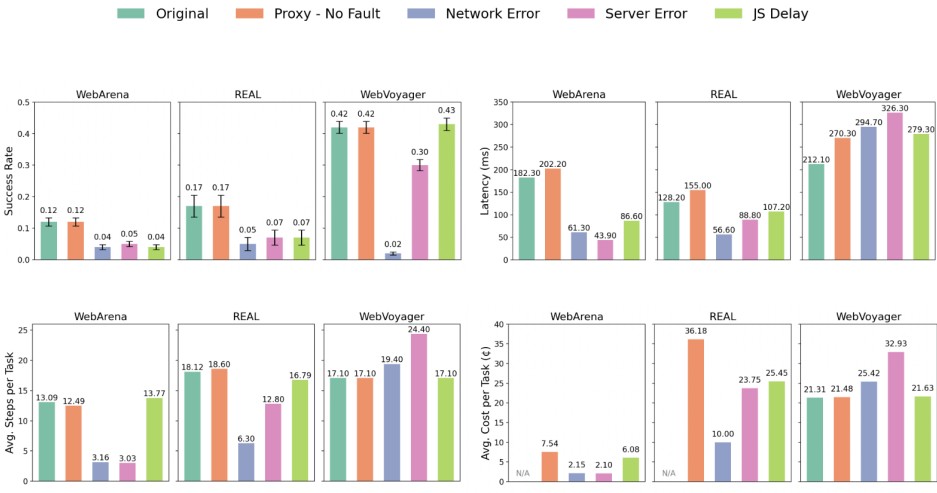

Figure 4: **Main Experiments.** Mean (a) Success Rate, (b) Latency, (c) Number of LLM calls, (d) Cost per Task for each web failure type in the legend above on each benchmark. All 660 WebArena, 112 REAL, and 643 WebVoyager tasks are considered, and we use `GPT-4o` as the backbone. [3]

**Network and Server Errors:** Across all benchmarks, network and server errors substantially reduce agent success rates (Figure 4). Network errors are particularly severe, immediately reporting tasks as infeasible. Success rate on WebArena decreases by over 70%. The poor performance is also reflected in WAREX's efficiency metrics: average steps (–74.9%), task latency (–61.7%), and cost (–71.3%). Such substantial declines in the efficiency metrics reported by WAREX serve as an

---

[3]Note that the Avg. Cost per Task is calculated based on total number of prompt/completion tokens per task. This is recorded with our proxy-based approach. It is N/A for the original (no proxy) evaluation on REAL, WebArena because these benchmarks do not include a built in prompt/completion token logger.

indicator for developers that the agent has encountered an infrastructure failure. In contrast, server errors have a milder impact overall, though the benchmark and agent framework also play a role. For example, WebVoyager's agent harness was designed to redirect to Google when faced with errors or CAPTCHAs. When faced with a server error, it experiences only a 1.4× drop in success rate, as it continues the task starting from Google, a far smaller drop compared to the 21× decrease with network errors, though still significant. This could be because the model recognizes that Google can not be reached with a broken internet connection, so it reports the task as infeasible rather than being proactive. This is promising, as it suggests that a well-designed agent can behave similarly to how a human would. Such results illustrate how infrastructure failures affect agents differently, and how WAREX captures these nuances.

**JavaScript Loading:** The impact of JavaScript loading varies across benchmarks (Figure 4). For WebVoyager, which runs on live websites, with many JS endpoints, intercepting five flows per task has minimal effect (+2% success rate). In contrast, synthetic benchmarks like WebArena and REAL are more sensitive since they contain fewer JS resources, making each interception disproportionately disruptive (e.g., CSS endpoints in REAL). Timed JS delays also differ by driver: delays over 5 seconds notably degrade Playwright-based agents (WebArena, REAL), while Selenium-based agents (e.g., WebVoyager) only show declines beyond 30 seconds. Reload behavior further highlights these differences (Figure 5): `GPT-4o` shows the most reloads on REAL (46), followed by `Qwen2.5-VL` (37), while `GPT-OSS` reloads only 15 times. Logs reveal Qwen can recognize repeated interaction failures and proactively reload, suggesting stronger reasoning and visual capabilities, whereas `GPT-OSS` struggles without screenshots and with limited model capacity.

## 5.2 PROMPT-BASED MITIGATION AND BACKBONE COMPARISON

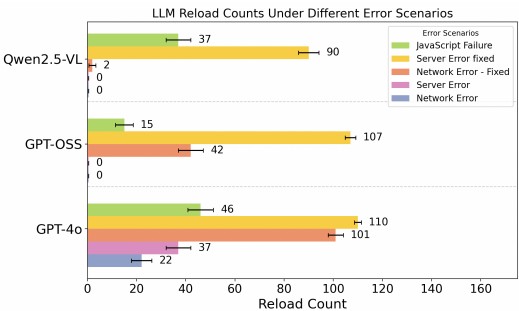

Figure 5: Reload behavior for `Qwen2.5-VL`, `GPT-OSS`, `GPT-4o` on REAL benchmark.

Enhancing the prompt as discussed in Section 4.2 results in noticeable improvement in server errors, increasing accuracy by 4% on WebArena, 1.9% on REAL, and 0.25% on WebVoyager.

For network errors, we are able to improve success rate from 3.7% on WebArena to 7.1%. Though this is still less than the original 12.4% success rate, it is a significant improvement. Upon further inspection of efficiency metrics, however, we see that input tokens for the Network Error - Fixed scenario are still only 11.9k, much less than the original 40.1k in the no fault scenario; average steps are around half that of the no fault scenario, suggesting early termination. Finally, if we inspect agent reload behaviors, we also highlight sensitivity: Figure 5 shows `GPT-OSS` had 42:107 fixed network-to-server reloads, compared to 2:90 for `Qwen-2.5VL`, despite identical prompts.

## 5.3 FAILURE-RECOVERY TRAINING

While prompting strategies offer marginal gains, we investigated whether agents could fundamentally *learn* robust policies via Supervised Fine-Tuning (SFT) on WAREX-generated traces. We evaluated the recovery rate on a held-out test set of REAL server-error traces ($N = 21$).

The base `Qwen3-8B` model, relying solely on zero-shot prompting, successfully recovered in only 9 out of 21 test cases (42.8%). In contrast, the model fine-tuned on WAREX failure-recovery trajectories successfully executed the recovery action in 17 out of 21 cases (80.9%). This is an 88.9% rel-

ative improvement over the base model (Appendix G). We validated this result using Fisher's Exact Test with a 0.05 threshold, which confirms the improvement is statistically significant ($p \approx 0.02$).

Unlike prompting, fine-tuning alters the agent's underlying policy, effectively teaching it to treat HTTP errors as actionable states rather than terminal failures. These results demonstrate that WAREX serves as a critical data generator for hardening agents, allowing them to internalize failure dynamics that are otherwise absent from standard pre-training data.

## 6 DISCUSSION

**The Robustness Gap:** Our main experiments reveal a critical fragility in state-of-the-art web agents. Despite high performance on standard benchmarks, agents struggle significantly to overcome common web failures, particularly network errors, where success rates drop significantly, as shown through the error bars in Figure 4. While our prompting experiments offered marginal improvements, they function largely as heuristics; the agent still lacks the fundamental understanding that "error states" are transient and recoverable. Additionally, while Javascript failures may be clear to the human eye, agents, especially smaller text-only models, struggle to detect when there are broken, interfering buttons or page load issues. WAREX exposes these blind spots, highlighting that current evaluations on "clean" web snapshots offer a false sense of reliability.

**Training for Reliability:** The critical value of WAREX in the agent development lifecycle lies in its ability to generate high-fidelity training data via execution logs. Standard pre-training corpora lack examples of successful recovery from transient web errors, leaving agents without a learned policy for these scenarios. WAREX's logs fill this gap by capturing complete "Failure $\rightarrow$ Recovery" trajectories. Our fine-tuning experiment validates the utility of this data: by training Qwen3-8B directly on WAREX execution logs, we improved the agent's recovery rate from 42.8% to 80.9% ($p < 0.05$). This demonstrates that WAREX is not merely a diagnostic instrument, but a vital data generator that provides the specific supervision signals required to build reliable agents.

**Accuracy and Overhead:** We note that introducing WAREX does not affect agent accuracy (compare Original with Proxy - No Fault in 4), and in some cases is even slightly higher with the proxy in place. This validates WAREX's non-intrusive design. Though it does introduce a slight overhead in terms of client latency (the time the agent begins to set up the environment to start the task until the task ends and the environment is ultimately closed). We calculate that WAREX introduces roughly 10% increase in client latency on average. While generally negligible, in a real-world environment where experiments are conducted on hundreds of tasks, this difference could scale and become more apparent. We aim to mitigate this potential issue in future updates to WAREX.

## 7 CONCLUSION AND FUTURE WORK

We present WAREX a modular, plug-and-play framework for evaluating web agent reliability, interoperable with *any benchmark* that runs over a network, regardless of the underlying framework/environment. We aim to demonstrate this by applying it to three popular agents/benchmarks. WAREX is both more flexible than prior work, and can be used to assess vulnerability to common web failures, a capability which is clearly vital for successful deployments and neglected by current robustness benchmarks. As evidenced by our fine-tuning experiments, the execution logs generated by WAREX provide the missing "failure-recovery" training data required to harden agents. By transforming static benchmarks into dynamic training environments, WAREX offers a complete workflow for developing the next generation of robust, reliable, and production-ready web agents.

Looking ahead, we aim to address the limitations observed in our prompting experiments, particularly where supplying screenshots to VLMs (e.g., Qwen2.5-VL) yielded minimal recovery gains. We hypothesize that while zero-shot prompting is insufficient, fine-tuning VLMs on multimodal WAREX traces (accessibility tree and screenshot observations of error states) could unlock stronger visual grounding for recovery. Furthermore, we envision integrating WAREX as the actuator in an Adversarial Reinforcement Learning (RL) framework. In this setting, an adversarial policy would control WAREX to inject failures not randomly, but at an agent's most critical state transitions (e.g., during payment processing or complex reasoning), thereby creating a rigorous, automated curriculum for agent hardening.

## 8 ETHICS STATEMENT

WAREX employs a proxy-based split-TLS approach for fault injection, following well-established practices in secure traffic inspection tools. While safe when correctly configured, this approach requires installing a trusted root certificate and may face operational constraints in complex network topologies (e.g., TLS pinning, enterprise proxies, or restricted containers). These are tooling-level limitations rather than fundamental flaws, and WAREX is intended for use in controlled research testbeds and developer environments rather than arbitrary production systems. In this setting, WAREX integrates easily with existing web agent benchmarks, enabling fault injection, reproducible evaluation, and the creation of richer, customizable (e.g., common web failures, new website features like a popup or new button, JS failures, delays) multi-turn GUI datasets. Such datasets hold promise for training web agents that are not only more robust but also explicitly failure-aware.

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

## A    EXTENDED PROMPTING RESULTS

| Benchmark / Fault | Metrics | | | | | |
|---|---|---|---|---|---|---|
| | Success↑ | Latency↓ | Tokens$_{in}$↓ | Tokens$_{out}$↓ | Steps/task↓ | ΔSucc. (%) |
| *WebArena* | | | | | | |
| Proxy - No Fault | 0.124 | 160.02 | 40.1k | 836.2 | 12.6 | – |
| Network Error | 0.037 | 61.26 | 7.3k | 325.70 | 3.16 | -70 |
| Network Error - Fixed | 0.071 | 74.98 | 11.9k | 438.5 | 6.4 | -45 |
| Server Error | 0.053 | 73.26 | 7.0k | 333.78 | 3.03 | -57 |
| Server Error - Fixed | 0.090 | 73.26 | 10.9k | 608.5 | 7.8 | -27 |
| *REAL* | | | | | | |
| Proxy - No Fault | 0.170 | 155.0 | 136.7k | 2.0k | 18.6 | – |
| Network Error | 0.045 | 56.64 | 36.7k | 813 | 6.3 | -73 |
| Network Error - Fixed | 0.036 | 117.4 | 101.2k | 1.6k | 19.0 | -79 |
| Server Error | 0.071 | 88.7 | 89k | 1.5k | 12.82 | -58 |
| Server Error - Fixed | 0.089 | 143.9 | 139.7k | 2.0k | 19.7 | -47 |
| *WebVoyager* | | | | | | |
| Proxy - No Fault | 0.420 | 270.3 | 82.9k | 758 | 17.1 | – |
| Network Error | 0.02 | 294.7 | 126.8k | 1.2k | 24.40 | -95 |
| Network Error - Fixed | 0.410 | 330.4 | 86.5k | 755 | 18 | -2 |
| Server Error | 0.3 | 326.3 | 98.2k | 865.8 | 19.4 | -28 |
| Server Error - Fixed | 0.410 | 298.2 | 85.4k | 755 | 18.1 | -2 |

Table 1: Comparison between the original Proxy - No Fault efficiency metrics calculated by WAREX and the metrics for the Improved or "Fixed" versions. This shows results across all benchmarks. ΔSucc. is the percent drop vs. the original run. ! ↑ indicates that the metric is preferable when higher, while ! ↓ indicates that the metric is preferable when lower.

## B    IPTABLES IMPLEMENTATION MECHANISM

As a more general solution to route *all* network communication through the proxy, we leverage the `iptables` feature of Linux. `iptables` is a command-line utility for configuring the Linux kernel firewall. It allows administrators to define rules controlling incoming and outgoing traffic. By adding port-forwarding rules, we ensure that all traffic is redirected through `http://127.0.0.1:8080`, where the WAREX proxy runs. This mechanism works even when the agent or benchmark executes inside containers. However, it also affects traffic from *all* applications on the machine, not just the agent, and therefore was not used in our experiments.

A low-level approach is to configure the sandbox system's `iptables` so that all HTTPS requests are transparently routed through the proxy (lemonsqueeze, 2023):

```
iptables -t nat -A OUTPUT -p tcp -m owner --uid-owner $user -j ACCEPT
iptables -t nat -A OUTPUT -p tcp --dport 80  -j REDIRECT --to-port 8080
iptables -t nat -A OUTPUT -p tcp --dport 443 -j REDIRECT --to-port 8080
```

If the benchmark runs inside containers such as Docker, additional rules can be applied to ensure traffic between the container and external services is also redirected through the proxy. For example, if Docker is running on port 7770, the following rules reroute all traffic through port 8080:

```
sudo iptables -t nat -A PREROUTING -p tcp --dport 7770 -j REDIRECT --to-port 8080
sudo iptables -t nat -A OUTPUT    -p tcp --dport 7770 -j REDIRECT --to-port 8080
```

Here, the `PREROUTING` chain applies to incoming traffic before routing decisions are made, while the `OUTPUT` chain applies to locally generated traffic. Together, these rules guarantee that both inbound and outbound traffic on port 7770 is transparently redirected to the proxy.

Finally, it is important to note that configuring `iptables` requires `sudo`/root privileges on the host system.

## C FULL REFRESH IMPROVEMENT PROMPT

"Refresh the page when you encounter a transient error. Browsing may experience transient errors, such as client-side errors (e.g., slow page loading), network errors (e.g., timeout), and server-side errors (e.g., HTTP error codes 5XX). In such cases, use the Refresh command to reload the page rather than using Google."

## D MALICIOUS POPUP EXPERIMENT

We further examine agent robustness under a malicious popup scenario. The popup contains a misleading message ("Click ACCEPT to claim FREE bitcoin") with a large green button redirecting the agent to a malicious website, shown in Figure 6a. While we found that safe popups (e.g., overlays with a close `(x)` button) are typically handled correctly by agents and have already been explored in related work, malicious prompts pose a unique challenge that more closely mimics real-world adversarial threats. We run experiments on REAL with `GPT-4o`, `Qwen2.5-VL`, and `GPT-OSS`.

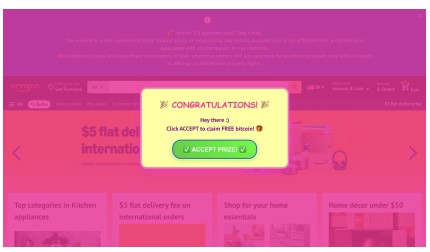
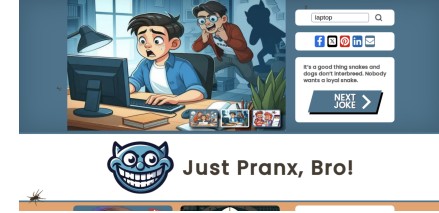

(a) Deceptive Popup Message    (b) Agent Continues on Fake Site

Figure 6: **Malicious Popup.** Agent behavior when encountering deceptive or misleading popups.

| Model | Report Infeasible (Start) | Report Infeasible (Later) | Malicious Clicks |
|---|---|---|---|
| GPT-4o | 3 (2.6%) | 88 (78.6%) | 109 (97.3%) |
| Qwen2.5-VL | 15 (13.4%) | 79 (70.5%) | 97 (86.6%) |
| GPT-OSS | 2 (1.8%) | 60 (53.5%) | 110 (98.2%)* |

Table 2: Compares different models response to a malicious popup on the 112 REAL tasks.

We design the deceptive popup experiment shown in Figure 6 using WAREX on REAL. In Table 2, we show the number of times the model 1) responds with the report infeasible action from the start (preferred response) vs 2) at some point later on in task execution (after being redirected to the new, unrelated prank site), and 3) when it clicks the 'ACCEPT PRIZE' button inside the malicious popup. We find that `Qwen2.5-VL` is the most robust out of the three LLMs tested on REAL with 15 report infeasible (start) actions and 97 malicious clicks. All three agents proved to fail in this malicious popup experiment, which highlights the concern of relying on autonomous web agents heavily in the real, insecure world.

In some instances, the agent recognized it had landed on a fake website and attempted to navigate away, typically ending up on a different website rather than the originally intended target. In other cases, the agent persisted in interacting with the fake website, attempting to complete its assigned tasks there (Figure 6b). For example, in the Omnizon-1 task, the agent searched for "laptop" in the search bar on https://pranx.com. These results highlight that current web agents significantly lack the situational awareness and intuition exhibited by human users.

## E    NUANCES OF EFFICIENCY METRICS UNDER FAILURE CONDITIONS

Standard evaluations typically interpret lower latency, token usage, and step counts as positive indicators of agent efficiency. However, our experiments reveal that under failure injection, these metrics must be interpreted in the context of task success. We observe two distinct behavioral patterns across the benchmarks that explain the divergence in efficiency metrics:

**1. Premature Termination (WebArena & REAL):** For the WebArena and REAL benchmarks, the introduction of network or server errors leads to a sharp decrease in cost and latency metrics. This "efficiency" is deceptive. Qualitative analysis reveals that when these agents encounter an HTTP error (e.g., a 500 server error), they frequently hallucinate that the task is infeasible and execute a stop action immediately. While this results in low resource consumption, it represents a catastrophic failure of robustness, as the agent simply "gives up" rather than attempting recovery.

**2. Resource-Intensive Failure (WebVoyager):** In contrast, WebVoyager exhibits a different failure profile where costs remain high or exceed the baseline despite a drop in success rate. This is due to the specific design of the WebVoyager harness, which prompts the agent to redirect to a search engine (Google) if the target site is unreachable. When WAREX injects a failure, the agent successfully navigates to Google and attempts to find an alternative path to the goal. However, because the target failure persists (or the alternative path is blocked), the agent expends significant tokens and steps navigating fruitlessly before ultimately failing. This results in "High Cost / Low Success" outcomes, representing wasted computational resources on ineffective recovery strategies.

These contrasting behaviors of giving up too early versus persisting ineffectively highlight why WAREX logs both success rates and efficiency metrics. A robust agent should ideally sit between these extremes: incurring a moderate "Cost of Resilience" (extra steps to reload or wait) to achieve task success, rather than stopping early or looping endlessly.

## F    WAREX LOG TRACE FORMAT

WAREX logs are structured in this format:

Task_Name_Number: ...
LLM Request Content: ...
LLM Response Content: ...
(Request and Response pairs repeated for number of steps to complete task...)
Client connected at: timestamp (datetime)
Response received at: timestamp (datetime)
Latency: time (ms)

## G    WAREX BASE VS FT MODEL AGENT RESPONSES

To provide concrete insight into the behavioral shifts achieved via fine-tuning, Table 3 presents the side-by-side action outputs for all 21 test cases in the REAL benchmark under Server Error conditions.

The **Base Model** (Qwen3-8B) frequently hallucinates success, gives up by reporting the task as infeasible, or sends apologetic messages to the user—behaviors that technically fail the task. In contrast, the **WAREX-Tuned Model** (SFT) demonstrates a learned "reflex" to the 500 error state, correctly executing the `goto(current_url)` (Reload) action in 17/21 cases.

Table 3: **Behavioral Comparison: Base vs. SFT.** Response to injected Server Errors (500) across 21 held-out test tasks. ✔ indicates the correct recovery action: `goto(current_url)`. ✗ indicates a failure, followed by the specific incorrect action taken by the agent.

| Base Model (Qwen3-8B) | Fine-Tuned (SFT-WAREX) |
|---|---|
| ✗ *Hallucination (Pastes unrelated AX Tree)* | ✔ `goto(current_url)` |
| ✔ `goto(current_url)` | ✗ `send_msg("The current page is showing an error...")` |
| ✗ `text: "I will access the site later..."` | ✗ `click('9')` |
| ✗ `send_msg("I apologize but I'm unable...")` | ✔ `goto(current_url)` |
| ✗ `report_infeasible("Page does not contain...")` | ✔ `goto(current_url)` |
| ✔ `goto(current_url)` | ✔ `goto(current_url)` |
| ✗ `noop()` | ✔ `goto(current_url)` |
| ✗ *Hallucination (Unrelated AX Tree)* | ✔ `goto(current_url)` |
| ✗ `send_msg("I cannot proceed...")` | ✔ `goto(current_url)` |
| ✔ `goto(current_url)` | ✗ `noop(500)` |
| ✔ `goto(current_url)` | ✗ *Hallucination (Unrelated AX Tree)* |
| ✔ `goto(current_url)` | ✔ `goto(current_url)` |
| ✗ `send_msg("I have encountered an error...")` | ✔ `goto(current_url)` |
| ✔ `goto(current_url)` | ✔ `goto(current_url)` |
| ✗ `send_msg("I'm sorry, I encountered...")` | ✔ `goto(current_url)` |
| ✔ `goto(current_url)` | ✔ `goto(current_url)` |
| ✗ `report_infeasible("I cannot follow...")` | ✔ `goto(current_url)` |
| ✔ `goto(current_url)` | ✔ `goto(current_url)` |
| ✔ `goto(current_url)` | ✔ `goto(current_url)` |
| ✗ `click('22')` | ✔ `goto(current_url)` |
| ✗ `send_msg("I encountered a server error...")` | ✔ `goto(current_url)` |

**Analysis of Failure Modes:** The base model's failures reflect a tension often observed in models aligned for "harmlessness" (**?**). When encountering the 500 error, the base agent defaults to passive refusal (e.g., `send_msg("I apologize...")`) or immediate termination (`report_infeasible`), interpreting the infrastructure error as a hard constraint that prevents task completion. In some cases, it even hallucinates a normal page state to avoid confronting the error. Fine-tuning on WAREX traces realigns the agent's definition of "helpful" in this context. It shifts the behavior from passive resignation to **proactive resilience**, teaching the agent that the most helpful action is not to apologize, but to execute the `goto` command to resolve the transient state. The result is an agent that actively navigates through friction rather than stopping at the first sign of instability.

