# OpenReview forum: "WAREX: Web Agent Reliability Evaluation on Existing Benchmarks"
_ICLR.cc/2026/Conference — Submitted to ICLR 2026_

### Official Review · Reviewer_cv5z · 2025-10-25

**Soundness:** 3
**Presentation:** 4
**Contribution:** 2
**Rating:** 6
**Confidence:** 3

**Summary:**

WAREX proposes a plug-and-play, proxy-based framework for evaluating reliability and adversarial robustness of browser-based LLM web agents by intercepting and modifying HTTP(S) traffic between the agent and web servers. WAREX uses a Mitmproxy addon + config.json that specifies failure types (network delays, server-side 4xx/5xx, JS loading failures, popups/overlays) and frequency rules (regex/exact URL match, k-th occurrence, random n times). It logs efficiency metrics (latency, API call counts, token usage) by intercepting outbound calls to LLM backends. The authors validate WAREX on three popular benchmarks (WebArena, REAL, WebVoyager) and three agents (SteP, REAL demo, WebVoyager agent) with multiple LLM backbones (GPT-4o, Qwen2.5-VL, GPT-OSS), showing substantial drops in task success under injected failures and limited mitigation from simple prompting.

**Strengths:**

- Practical, reproducible engineering: Uses Mitmproxy and provides clear setup options (env vars, Playwright proxy, iptables) so replication is straightforward.
- Broad empirical evaluation: Tests across three benchmarks, three agents, and multiple LLMs; measures both success and efficiency metrics (tokens, latency). This breadth strengthens generality claims.
- Novel framing: Shifts attention from purely adversarial overlay attacks to common web failures (network/server/JS), which are practically important and understudied.
- Non-intrusive instrumentation + logging: Ability to capture LLM call metadata and token counts even for closed-source benchmarks is valuable for auditing.

**Weaknesses:**

Covered in questions

**Questions:**

- What exactly is logged for “LLM interactions”? Do you log full prompts and responses, or only token counts and timestamps?
- How did you choose the default injection magnitudes (10s delay, error rates, frequency rules)? Can you provide evidence these match real-world distributions (or include a sensitivity sweep)?
- Mitigation attempts beyond prompting: Did you try non-prompting mitigations (retry heuristics in harness, DOM-based checks, or simple rule-based filters) and, if so, how did they compare? If not, can you add a small evaluation?
- Please provide example per-step traces (timestamped) for at least 3 tasks: a successful baseline, a network-failure run where the agent fails, and the malicious-popup scenario where it clicks. If full prompts cannot be released, provide redacted prompt snippets and token counts to understand what’s going under the hood.

---

> ### Author Response · Authors · 2025-11-29
>
> We thank the reviewer for the encouraging assessment of WAREX’s practical engineering value. We appreciate your detailed technical questions regarding our implementation choices and mitigation strategies.
>
> **Q1: What exactly is logged for “LLM interactions”? Do you log full prompts and responses, or only token counts and timestamps?**
>
> We log the full HTTP request and response bodies (including the full prompt context window and the generated output), alongside token counts and timestamps.
> * Metric Calculation: This logging allows us to precisely calculate the "Cost per Task" and "Latency" metrics reported in Figure 4.
> * Training Data: Capturing the full interaction history allowed us to construct the training dataset for the new fine-tuning experiment described in Section 4.2, effectively turning WAREX from a diagnostic tool into a dataset generator.
>
> **Q2: How did you choose the default injection magnitudes (10s delay, error rates, frequency rules)? Can you provide evidence these match real-world distributions (or include a sensitivity sweep)?**
>
> * Delay: Our choice of a 10s delay was an engineering decision to balance experimental throughput with validity. While standard browser page load timeouts often default to 30s, waiting this full duration for every injection would have drastically increased benchmark runtime, making large-scale data generation prohibitively slow. 10s served as an optimal "soft timeout" that effectively disrupted the agent's execution loop without incurring the massive latency overhead of a full driver timeout.
> * Failure Frequencies: As detailed in Section 3.3, WAREX supports both deterministic injection (e.g., consistent first-page failures for reproducibility) and stochastic injection (randomized occurrences). This flexibility enables developers to cover the full evaluation spectrum, ranging from targeted debugging to generalized robustness testing against unpredictable infrastructure instability. We’ve simplified this by defining the failure frequency parameter: x = 0 triggers injection on the first page, x > 0 injects failures randomly *x* times, and x < 0 disables injection. These details are explained in Section 3.3.
>
> **Q3: Mitigation attempts beyond prompting: Did you try non-prompting mitigations (retry heuristics in harness, DOM-based checks, or simple rule-based filters) and, if so, how did they compare? If not, can you add a small evaluation?**
>
> We performed a new experiment to demonstrate that WAREX functions as a generator of Failure-Recovery Trajectories (detailed in the updated Section 4.2).
> * Experimental Design: We fine-tuned (SFT) Qwen3-8B-Base using PEFT with LoRA on traces generated by WAREX over the REAL benchmark. We focused on server errors as the expected recovery action is the refresh/goto(same_url) action, and we talk about this extensively in the paper. We generated traces for all 112 tasks on REAL, and separated the 11 benchmarks into train (9), val (1), test (1). Out of these 112 tasks, we took a balanced set of server and non-server error - half server and half non-server error - traces resulting in train (total 244 traces), val (42), test sets (42).
> * Results: We measure specifically the number of proper reloads for the server error pages (goto(current_url) used). The base model successfully recovered from server errors only 42.8% (9/21) of the time. The WAREX-tuned model improved this to 80.9% (17/21). This is an 88.9% relative improvement in reliability.
> * Conclusion: This proves WAREX provides unique training signals. Standard web data and benchmarks filter out errors, while WAREX re-introduces them as "learning moments," allowing agents to learn policy adjustments (robustness).
>
> **Q4: Please provide example per-step traces (timestamped) for at least 3 tasks: a successful baseline, a network-failure run where the agent fails, and the malicious-popup scenario where it clicks. If full prompts cannot be released, provide redacted prompt snippets and token counts to understand what’s going under the hood.**
>
> We have added a new Appendix section containing the structure of each log file. We would like to note that we moved the malicious popup scenario to Appendix D to include the more crucial failure-recovery training experiment. Due to the substantial size of the full execution logs (which contain complete prompt contexts and entire accessibility tree observations), we include an example log for a task on REAL for each error type discussed in the supplementary code submission under the log_traces folder. In Appendix F, we provide the schema used to structure these logs.

---

### Official Review · Reviewer_qrMr · 2025-10-30

**Soundness:** 3
**Presentation:** 2
**Contribution:** 3
**Rating:** 2
**Confidence:** 4

**Summary:**

The paper introduces WAREX, a framework designed to evaluate the reliability of LLM-based web agents under realistic, failure-prone web conditions. Unlike existing benchmarks that assume stable and deterministic websites, WAREX functions as a network-layer proxy that injects real-world issues such as network delays, server-side errors, JavaScript failures, and malicious popups into existing benchmarks without requiring modifications to the agents or their environments. The framework’s main contribution is to provide a plug-and-play mechanism for simulating diverse web failures and adversarial scenarios across benchmarks, while also logging efficiency metrics such as latency, API call counts, and token usage to jointly assess robustness and computational cost.

Through experiments on WebArena, REAL, and WebVoyager, the authors show that current state-of-the-art web agents using models like GPT-4o, Qwen2.5-VL, and GPT-OSS experience substantial performance degradation when subjected to such injected faults, with success rates decreasing by up to 70–95%. The study also finds that most agents remain vulnerable to deceptive popups, and that prompt-based mitigation strategies only partially alleviate these issues. Overall, WAREX provides a benchmark-agnostic and extensible methodology for evaluating the robustness, safety, and efficiency of web agents, aiming to bridge the gap between controlled benchmarks and real-world deployment conditions.

**Strengths:**

The paper addresses an important and timely problem by introducing a benchmark framework that tests web agents under realistic and failure-prone conditions. This type of reliability evaluation is currently missing in the field, and WAREX provides a practical and well-motivated step toward bridging the gap between controlled benchmarks and real-world deployment.

**Weaknesses:**

The experimental sections are difficult to follow, which raises uncertainty about the reliability of the results. In particular, there are no uncertainty or variance metrics, making it impossible to judge whether the reported differences are statistically significant. Section 4.2 is also awkwardly structured: it introduces experiments but defers their explanation to Section 5, which breaks the logical flow and should be merged. Overall, the presentation of the experiments is not very clear or polished, which makes it harder to assess the validity of the findings.

The figures and tables could be improved. It is not obvious which agent is represented in Figure 4, despite the manuscript stating that three agents were tested across three benchmarks. Contrasting Figure 4 and Table 5 is confusing and should be avoided; the results should instead be presented consistently in one place, with deltas or comparisons made directly in the text or tables. In Figure 6, two of the three models appear to be missing data columns, which leaves the comparison incomplete.

The efficiency metric is also problematic. Metrics such as “average cost per task,” “latency,” and “cost” all decrease under failure conditions, which the paper interprets as negative outcomes. However, lower values for those metrics are nominally positive. The evaluation should be redesigned to emphasize high performance together with low cost, latency, and number of steps, rather than treating lower values as inherently bad.

Other presentation issues include the absence of references to Table 1, weak figure captions that do not make figures self-contained, and numerous missing citations, especially in the related-work and methods sections. Overall, the paper would benefit from a more coherent and polished presentation of results and supporting materials

**Questions:**

Missing citation to BrowserGym (De Chezelles et al., 2024) when mentioning prior web-agent environments, as it provides a unified gym-style framework.

The claim about the lack of real-world web benchmarks should acknowledge WorkArena (Drouin et al. 2024)  and WorkArena++ (Boisvert et al., 2024), which specifically target realistic enterprise web tasks. F

inally, AgentRewardBench (Lù et al., 2025) should be cited for its analysis of side effects and looping behaviour in agent evaluation.

---

> ### Author Response · Authors · 2025-11-29
>
> **Q1: Missing Citations**
>
> We thank the reviewer for pointing out these missing citations. We have added a reference to BrowserGym (Le Sellier De Chezelles et al., 2025) in Section 2.
>
> WorkArena (Drouin et al., 2024) and WorkArena++ (Boisvert et al., 2025b), now mentioned in Section 1, provide valuable benchmarks focused on realistic enterprise-oriented tasks, such as working with lists, forms, dashboards, and we agree that strong web agents should ultimately be able to perform these tasks. Our work, however, aims to evaluate robustness under controlled failure injection, which requires broad coverage across diverse website structures and interaction patterns. For this reason, we selected WebArena (popular benchmark, which offers diverse but controlled simulated environments), REAL (newer benchmark with realistic website clones built with React and Next.js), and WebVoyager (tasks on live, real-world websites). This combination allows us to test agent recovery behavior across controlled environments, synthetic clones, and actual production websites.
>
> We have also included AgentRewardBench (Lù et al., 2025) in Section 1 as a new relevant benchmark.
>
> **Experimental Structure**
>
> We have significantly restructured the Experimental Design (Section 4) and Results (Section 5) sections to unify the presentation and improve logical flow. The revised paper is now organized into three distinct experimental phases, consistent across both the methodology and results sections:
>
> In Section 4.2, we detail the specific  three primary experimental phases:
> * 1) Main Robustness Evaluation (GPT-4o): We conducted a comprehensive analysis across all three benchmarks using GPT-4o. This involved seven distinct configurations, each utilizing a single fault injection on the initial page: (1) No Proxy (original baseline), (2) Proxy Without Faults (infrastructure control), (3) Network Error, (4) Server Error, (5) JavaScript Delay, (6) Network Error with Prompt Mitigation, and (7) Server Error with Prompt Mitigation.
> * 2) Prompt-Based Mitigation and Backbone Comparison: To assess generalization, we extended the prompt-based mitigation experiments to diverse model architectures. We evaluated Qwen2.5-VL, GPT-4o, and GPT-OSS against the various failure types to determine how different backbones respond to standard mitigation strategies.
> * 3) Failure-Recovery Training: We performed a fine-tuning experiment using Qwen3-8B on traces generated from the REAL benchmark. This experiment demonstrates that WAREX functions beyond a simple diagnostic tool; its execution logs serve as training data that can be used to actively fix bugs and improve agent reliability.
>
> **Figure Clarity**
> For the bars in Figure 5 that did not appear, this occurred because they had a value of 0 (e.g., 0 server errors reloaded with GPT-OSS). We have updated the figure to make it clear that each bar is present and to ensure that their colors are visible (see Figure 5).
>
> We clarify that in Figure 4, "Agent" refers to the specific agent architecture designed for that benchmark (e.g., the "SteP" agent for WebArena, the demo agent for REAL, the "WebVoyager" agent for WebVoyager), all utilizing the same GPT-4o backbone to ensure fair comparison. We have updated the Experimental Setup (Section 4.1) to state more clearly which agent was tested on which benchmark.
>
> To improve readability, we have relocated the comprehensive prompt-based mitigation results in Table 1 to Appendix A, as mentioned in Section 4.2. We retained Figure 5 (formerly Figure 6) in the main text, as it efficiently synthesizes three key experimental dimensions: the new recovery rate metric, the impact of prompting strategies, and performance comparisons across different LLM backbones.
>
> **Showing Statistical Significance**
>
> We added standard error rates to Figures 4 and 5 to make it more clear that the results are statistically significant. In Figure 4, one can see that the server and network error bars do not overlap with the original/no proxy bars for the success rate experiments. Additionally, for the new failure-recovery training experiments, we have validated our results using Fisher's Exact Test as shown in Section 5.3.
>
> **Nuances of Efficiency Metrics**
>
> We agree that lower cost/latency is nominally positive; however, our analysis reveals that under failure injection, these metrics often serve as proxies for behavioral fragility rather than true efficiency.
>
> We have added a new section (Appendix E) to thoroughly address this. In short, we distinguish between two failure modes exposed by WAREX:
> * Premature Termination (WebArena/REAL): Where agents "give up" immediately upon error, leading to deceptively low costs.
> * Ineffective Persistence (WebVoyager): Where agents redirect (e.g., to Google) and burn tokens attempting to recover without success, leading to high costs.
>
> We believe this new appendix provides the necessary context to further understand the experimental results and the benefits of WAREX.

---

### Official Review · Reviewer_D6ms · 2025-10-30

**Soundness:** 2
**Presentation:** 2
**Contribution:** 1
**Rating:** 2
**Confidence:** 4

**Summary:**

The paper introduces WAREX, a proxy sitting between a web-agent and a benchmark environment. The proxy rewrites HTTP responses to simulate web errors and adversarial conditions. The authors argue that injection at the network layer makes WAREX benchmark and agent agnostic. They run a GPT-4o-based agent on three existing web benchmarks with WAREX and show that performance decreases when errors are injected, and that some of it can be recovered by prompting.

**Strengths:**

- Demonstrates the brittleness of (some) agents under common web errors.
- Gives a benchmark-agnostic way to inject failures via network interception.

**Weaknesses:**

- Lacks novelty: differences from prior work (e.g. DoomArena, 2025) are primarily engineering rather than scientific, changing the attachment point to be the network rather than wrapping the agent or env.
- Weak backbone models: GPT-4o, Qwen2.5, and GPT-OSS are relatively poor models for agentic tasks compared to current SoTA. WebArena and REAL success rates reported (without errors) are far below current reported performance for strong agents.
- Shallow analysis: the paper only shows drops in performance and reports behaviour, without a real exploration of the cause of failures.
- Results unsurprising: if there is an error, and the agent doesn't know to reload, it will tautologically fail.
- REAL already allows configuring errors and latency, and so the benefit of WAREX in this case is unclear.
- Difficulty of comparison: to allow future work to compare fairly, a record of the faults injected and when would be necessary.

**Questions:**

- It would strengthen the paper a lot to show that training on successful WAREX traces, or some sort of WAREX-agent improves robustness across failure modes (including OOD ones). Could this be done?
- The picture referred to as Omnizon homepage looks like a calendar website, whereas you state that it is an Amazon clone. Which is it?
- Can you think of a better metric to capture an agent's recovery ability than simply 0-1 overall task success?

---

> ### Author Response · Authors · 2025-11-29
>
> We thank the reviewer for their assessment. We would like to demonstrate that WAREX is not just a tool for breaking agents, but a necessary methodology for fixing them.
>
> **Q1: It would strengthen the paper a lot to show that training on successful WAREX traces, or some sort of WAREX-agent improves robustness across failure modes (including OOD ones). Could this be done?**
>
> We performed a new experiment to demonstrate that WAREX functions as a generator of Failure-Recovery Trajectories. As detailed in the updated Section 4.3 ("Failure-Recovery Training"), we fine-tuned Qwen3-8B using LoRA on 244 execution traces generated by WAREX on the REAL benchmark. These traces specifically captured trajectories where an injected server error was followed by a correct recovery action (page reload). The fine-tuned agent achieved a 17/21 (80.9%) recovery rate on a held-out test set, compared to only 9/21 (42.8%) for the base model, an ~89% relative improvement.
>
> OOD Errors: We noted that WAREX can introduce random or out-of-distribution (OOD) errors rather than only intercepting the first page. In our new fine-tuning experiments, we inject errors on random (OOD) pages so the agent cannot rely on refreshing on the first page and instead must learn genuine recovery behaviors.
>
> **Q2: The picture referred to as Omnizon homepage looks like a calendar website, whereas you state that it is an Amazon clone. Which is it?**
>
> We appreciate the reviewer for catching this typo and have corrected the label in Figure 2 to reflect that the image is from the GoCalendar website, not Omnizon.
>
> **Q3: Can you think of a better metric to capture an agent's recovery ability than simply 0-1 overall task success?**
>
> In our Prompt-Based Mitigation and Backbone Comparison experiment results (5.2), we report the number of times the agent executes a recovery action (e.g., reload) in response to injected server, network, or JavaScript errors.
>
> If we express WAREX’s logging structure as (State) &rarr; (Injected Failure) &rarr; (Recovery Action), an intuitive recovery metric is: **Recovery Rate = (Number of times a correct recovery action was taken) / (total number of injected failures)**. This is described in more detail in our updated Section 4.2.
>
> **Novelty of WAREX**
>
> We have expanded Section 2 to demonstrate WAREX's novelty.
> * **Application-level vs. Network-Level (WAREX)**: DoomArena works at the application level, where the authors inherit the environment from the benchmark and use a designed AttackConfig to modify the observation before it goes to the agent which decides the next action. WAREX, on the other hand, operates at the network level, where a man-in-the-middle proxy used to genuinely modify the observation and server-side code, not just make it look like something is there which is not. This lower-level approach allows for *higher adaptability*.
>
> **Model Choice**
>
> * GPT-4o: Primary closed-source baseline used as LLM backbone in most SoTA web agents.
> * Qwen2.5-VL: Vision-based open-source baseline to test whether visual grounding improves reliability.
> * GPT-OSS-20b: Open-weight text model (released August 2025), ensuring accessible, reproducible comparisons.
>
> **Failure Modes**
>
> Our results show that base models do not exhibit tautological failures (e.g., solely crashing); instead, they display highly stochastic and sometimes deceptive behaviors (Appendix E). While such variability may be useful in creative tasks, reliable agents require stable, deterministic recovery. WAREX exposes this lack of stability and enables measurement through the Recovery Rate defined in Section 4.2.
>
> **REAL Error Configurations**
>
> While REAL supports internal configuration, its error/latency settings are applied at the application level, not the network-level. End-users of REAL cannot specify custom error injections. Increased latency or errors are added globally to *all* page loads. WAREX, on the other hand, enables precise control over URL patterns and types of failures, giving researchers the ability to inject specific error modes only at particular pages or interactions. This supports more controlled, realistic studies of agent robustness and error recovery.
>
> **Fault Injections**
>
> In Section 4.2, we detail the specific failure modes and frequencies employed. For our main experiments, we explored the following seven distinct configurations, each utilizing a single fault injection on the initial page: (1) No Proxy (original baseline), (2) Proxy Without Faults (infrastructure control), (3) Network Error, (4) Server Error, (5) JavaScript Delay, (6) Network Error with Prompt Mitigation, and (7) Server Error with Prompt Mitigation. We also explored malicious popup attacks in Appendix D.

---

### Official Review · Reviewer_zE6V · 2025-11-01

**Soundness:** 3
**Presentation:** 3
**Contribution:** 2
**Rating:** 6
**Confidence:** 3

**Summary:**

WAREX is a plug-and-play framework designed to augment existing web agent benchmarks (such as WebArena, REAL, and WebVoyager) with realistic stress conditions. Existing evaluations rely on controlled environments, giving a false sense of reliability. WAREX acts as a transparent proxy layer by intercepting and modifying HTTP(S) traffic to inject common web failures (e.g., network delays, server errors) and adversarial attacks (malicious popups). Experiments show that introducing WAREX leads to significant drops in task success rates (e.g., over 70% decrease due to network errors on WebArena), exposing fundamental robustness gaps in state-of-the-art agents.

**Strengths:**

The primary strength of WAREX is its maximum interoperability, operating at the network layer via a transparent proxy. This modular design is benchmark-agnostic and requires no changes to agent or benchmark source code, facilitating its use with existing, even closed-source, resources. WAREX systematically evaluates robustness against pervasive, everyday web failures (network, server, JavaScript errors), beyond the narrow focus of previous work on overlays. It also offers a dual capability by logging efficiency metrics, such as LLM token counts and latency, providing a holistic reliability and cost analysis.

**Weaknesses:**

The design philosophy and measurement results of the "stress test" proposed in this work for GUI Agents constitute an excellent engineering achievement. However, it needs to be demonstrated whether this can inform the Agent's capability training itself in a sufficiently profound and intuitive way, rather than merely being a superb engineering tool.

**Questions:**

1. Please explicitly state how the WAREX framework provides unique insights or data structures for training a new generation of failure-aware web agents, distinct from simple data augmentation or reinforcement learning environment setups.
2. What is the scalability of this framework? What are the next directions for expansion?

---

> ### Author Response · Authors · 2025-11-29
>
> We thank the reviewer for the constructive feedback and for recognizing WAREX as an "excellent engineering achievement." We appreciate the opportunity to clarify how WAREX is more than just a tool to find agent weaknesses in web failure scenarios, but can be incorporated as a core component of the agent development and training pipeline.
>
> **Q1: Please explicitly state how the WAREX framework provides unique insights or data structures for training a new generation of failure-aware web agents, distinct from simple data augmentation or reinforcement learning environment setups.**
>
> Unlike simple data augmentation, which typically applies random, unstructured noise, WAREX allows developers to target specific, consistent error distributions. It produces structured Failure-Recovery Trajectories: (State) &rarr; (Injected Failure) &rarr; (Recovery Action). Furthermore, WAREX resolves the scalability bottleneck inherent in standard Reinforcement Learning (RL) setups. In a traditional RL context, injecting these failures would require manually modifying the source code and reward functions for each individual benchmark environment. This approach is brittle and impossible to apply to closed-source benchmarks or live websites. By operating at the network layer rather than the application layer, WAREX decouples the failure injection from the environment logic. This allows it to function universally across any benchmark without requiring access to the underlying environment code.
>
> We performed a new experiment to demonstrate that WAREX functions as a generator of Failure-Recovery Trajectories (detailed in the updated Section 4.2).
> * Experimental Design: We fine-tuned (SFT) Qwen3-8B-Base using PEFT with LoRA on traces generated by WAREX over the REAL benchmark. We focused on server errors as the expected recovery action is the refresh/goto(same_url) action, and we talk about this extensively in the paper. We generated traces for all 112 tasks on REAL, and separated the 11 benchmarks into train (9), val (1), test (1). Out of these 112 tasks, we took a balanced set of server and non-server error - half server and half non-server error - traces resulting in train (total 244 traces), val (42), test sets (42).
> * Results: We measure specifically the number of proper reloads for the server error pages (goto(current_url) used). The base model successfully recovered from server errors only 42.8% (9/21) of the time. The WAREX-tuned model improved this to 80.9% (17/21). This is an 88.9% relative improvement in reliability.
> * Conclusion: This proves WAREX provides unique training signals. Standard web data and benchmarks filter out errors, while WAREX re-introduces them as "learning moments," allowing agents to learn policy adjustments (robustness).
>
> **Q2: What is the scalability of this framework? What are the next directions for expansion?**
>
> Our framework scales along two dimensions:
> * Across websites and logs:
> Since we use mitmproxy’s addon mechanism with domain/URL-based matching, we can easily support many websites by adding additional regex rules and routing each domain to its own log file.
> * Across machines (horizontal scaling):
> WAREX is stateless across tasks: each proxy instance only processes the flows it sees. Experiments can run in parallel. Users can run multiple agent and proxy servers. For example, experiments for each website in REAL could be handled on separate servers, and their logs could be aggregated afterward. Adding more servers linearly increases throughput.
>
> Next directions:
> * We plan to expand WAREX by developing new addons that (1) introduce additional failure modes (e.g., CAPTCHAs) and (2) incorporate more URL patterns to support new benchmarks and live sites, thereby improving research on generalizable agent creation.

---

### Author Response · Authors · 2025-11-29

## Summary of Changes

* **Added Failure-Recovery Training (Sections 4.2, 5.3)**:
Because 3 of the 4 reviewers explicitly asked for evidence that training on WAREX logs can improve agent robustness, we conducted a new fine-tuning experiment. This demonstrates that WAREX is not merely a diagnostic tool, but can be incorporated into the agent training process to develop reliable web agents.

* **Clarified Architectural Novelty (Section 2)**: We strengthened the distinction between WAREX’s Network Layer design and prior Application Layer approaches. We emphasize that by decoupling failure injection from the environment, WAREX achieves universal compatibility with closed-source benchmarks and live websites, capabilities that current wrapper-based methods lack.

* **Defined Novel “Recovery Rate” Metric (Section 4.2)**:
We formalized the first metric designed to isolate agent reliability under transient web failures. While existing benchmarks focus solely on accuracy, *Recovery Rate* quantifies an agent's specific policy robustness (e.g., reloading), filling a critical gap in failure-aware evaluation.

* **Restructured for Clarity and Rigor (Section 4, 5)**:
In response to Reviewer 3, we aligned the structure of the Experimental Design and Results sections for a cohesive narrative. We added standard error bars to all plots, and conducted Fisher’s Exact Tests for fine-tuning comparisons.

**Future Outlook:**
Per Reviewer 1's request for expansion directions, we have outlined plans to use WAREX for Adversarial RL curricula and for multimodal fine-tuning to address the severe visual grounding limitations we observed in standard VLMs.

---

### Meta-Review · Area_Chair_BGMi · 2025-12-31

**Summary:**

This submission proposes WAREX, a proxy-based framework for injecting web failures into existing web-agent benchmarks to evaluate robustness under more realistic conditions. While the problem is timely and the engineering effort appears substantial, the overall paper is better suited for reject at this stage given concerns raised across reviews regarding novelty, clarity, and experimental rigor.

The reviewers’ original average score is 4, and it is likely to remain largely unchanged after rebuttal, since several key concerns appear only partially addressed rather than fully resolved. In particular, the work still reads as relatively early-stage: the writing and experimental presentation need significant polishing, and the narrative around what is fundamentally new (beyond an engineering change in injection layer) remains insufficiently convincing. Claims around scalability and broader applicability would benefit from more concrete evidence and clearer articulation, rather than relying on high-level future directions.

Although the authors report some additions (e.g., limited extra experiments and reframing around recovery), multiple reviewer-suggested improvements either remain limited in scope or are deferred to “future work,” and therefore do not yet substantively change the overall assessment. Overall, the submission requires further development, especially clearer positioning of novelty, stronger and more convincing experimental design/analysis, and improved presentation, to reach the standard for acceptance, and therefore it remains in a reject state at present.

**Reviewer Concerns:**

Reviewers’ main concerns center on whether the work is sufficiently novel (vs. mostly an engineering change in where failures are injected) and whether the experimental baselines/models are strong enough to support the paper’s claims. They also criticize the experimental presentation and analysis as hard to follow or too shallow, with limited root-cause investigation beyond showing performance drops under injected faults. Several reviewers request better statistical treatment/uncertainty reporting, clearer figures/tables, and more appropriate metrics for “recovery” beyond overall task success, plus clearer interpretation of efficiency metrics like cost/latency. Finally, they ask for stronger evidence that WAREX logs can enable training or mitigation (not just diagnosis), along with better documentation of injection settings, logging details, and reproducible fault records/traces.

**Reviewer Scores:**

Probably all 4 reviewers will not change the original scores (no replies from reviewers during the discussion). Below are the detailed reasons.

- Reviewer zE6V (stays at 6)
This reviewer already viewed WAREX as a strong engineering contribution with high interoperability, but questioned whether it meaningfully informs training rather than just diagnosing failures.
The authors add a failure-recovery fine-tuning result and position WAREX logs as “Failure-Recovery Trajectories,” yet the reviewer may still keep a 6 because the paper’s contribution is still framed as “fair” and the new training evidence is limited to a specific setup (e.g., a particular error type / dataset design), leaving the broader “profound and intuitive” training impact not fully settled.

- Reviewer D6ms (stays at 2)
This reviewer’s rejection is driven by fundamental beliefs: novelty is mainly engineering (network-layer attachment vs wrappers), the backbone/agents are not strong enough for the claims, and the analysis is shallow/unsurprising (errors cause failure if the agent does not recover). The rebuttal introduces a recovery metric and a training experiment and clarifies some issues (e.g., figure typo, novelty discussion, REAL vs WAREX positioning), but these do not change the reviewer’s core premise that the work still lacks sufficient scientific novelty and compelling baseline strength, so probably the rating remains 2.

- Reviewer qrMr (stays at 2)
This reviewer’s main concern is that the experimental presentation is hard to follow and lacks uncertainty/statistical grounding, and they also object to confusing figures/tables and problematic interpretation of efficiency metrics. The authors claim to have restructured Sections 4–5, added standard errors and significance testing, and improved citations/figure clarity, but after reading, I think the paper's writing could be further improved. The reviewer likely still keeps a 2 as these changes during rebuttal time do not fully fix the underlying clarity and evaluation-design issues (especially the efficiency-metric framing).

- Reviewer cv5z (stays at 6)
This reviewer is broadly positive on practicality and breadth, while asking detailed questions about what is logged, why injection magnitudes were chosen, and whether mitigations beyond prompting were evaluated, plus requests for concrete per-step traces. The authors answer logging questions (full HTTP request/response bodies including prompts/outputs, plus token counts/timestamps) and point to appended log structure/examples, but the score likely remains 6 because the injection-parameter justification is still largely an engineering choice and the mitigation story remains limited (e.g., shifting to fine-tuning evidence rather than comparing simple non-prompt heuristics), so the paper improves but not enough to warrant a higher score.

---

### Decision · Program_Chairs · 2026-01-26

Reject